# Healthy Lifestyle Management of Pediatric Obesity with a Hybrid System of Customized Mobile Technology: The PediaFit Pilot Project

**DOI:** 10.3390/nu13020631

**Published:** 2021-02-16

**Authors:** Anna Pia Delli Bovi, Giorgia Manco Cesari, Maria Chiara Rocco, Laura Di Michele, Ida Rimauro, Anna Lugiero, Silvia Mottola, Anna Giulia Elena De Anseris, Lucia Nazzaro, Grazia Massa, Pietro Vajro

**Affiliations:** 1Pediatrics Section, Department of Medicine, Surgery and Dentistry “Scuola Medica Salernitana”, University of Salerno, 84081 Baronissi, Italy; delliboviannapia@gmail.com (A.P.D.B.); mancocesarigiorgia@gmail.com (G.M.C.); mariachiara@live.it (M.C.R.); lauradimichele93@gmail.com (L.D.M.); idarimauro93@gmail.com (I.R.); annalugiero95@gmail.com (A.L.); silvia_mottola_@hotmail.it (S.M.); graziamassa39@gmail.com (G.M.); 2Pediatrics/Clinical Pediatrics, “San Giovanni di Dio e Ruggi d’Aragona” University Hospital, 84131 Salerno, Italy; annagiulia.deanseris@gmail.com (A.G.E.D.A.); nazzaroluci@gmail.com (L.N.)

**Keywords:** obesity, lifestyle, dropout, mobile technology, attrition, pediatric

## Abstract

Pediatric obesity management strategies suffer from a high rate of dropout and persistence of weight excess, despite the use of new tools, such as automated mobile technology (MT). We aimed to compare the efficacy of two 6-month personalized MT protocols in terms of better engagement, adherence to follow-up visits and improved anthropometric and lifestyle parameters. MT contacts consisted of three personalized/not automated What’s App^®^ self-monitoring or challenge messages per week. Messages, sent by a dedicated coach were inserted between three-monthly in-presence regular visits with (PediaFit 1.2) or without (PediaFit 1.1) monthly free-of charge short recall visits carried out by a specialized pediatric team. The sample included 103 children (mean age 10 years, range 6–14) recruited in the Pediatric Obesity Clinic between January 2017 and February 2019, randomized into Intervention group (IG) (*n* = 24 PediaFit 1.1; *n* = 30 PediaFit 1.2) and Control group (CG) (total *n* = 49). Controls received standard treatment only (indications for healthy nutrition and physical activity, and three months in presence regular visits). Overall, both IGs achieved significantly better results than the CGs for all considered parameters. Comparison of the two IGs at the sixth month in particular showed an IG 1.2 statistically significantly lower drop-out rate (10% vs. 62%, *p* = 0.00009), along with significantly improved BMI (*p* = 0.003), Screen Time (*p* = 0.04) and fruit and vegetables consumption (*p* = 0.02). The study suggests that the hybrid association of messaging through personalized/not automated MT plus monthly free-of charge recall visits may improve the prefixed outcomes of MT weight loss intervention programs.

## 1. Introduction

Childhood obesity is a major public health problem that increases the risk of medical comorbidities, health care costs and decreased quality of life. The above factors underline the need for early actions [1]. However, these are often compromised by a high rate of dropout (i.e., abandoning the intervention before reaching the set goals) which may affect from 12%–50%, especially among pediatric participants [2].

Studies suggest that age, ethnicity, anthropometry, health service factors, and lack of treatment preparedness are likely predictors of dropout rates from weight management interventions, together with psychological distress and lower family functioning [3,4]. The latter aspect implies the importance of the household environment and of ongoing support for parents in an empathetic and personalized way rather than simply focusing on weight changes [3]. Intensity of the intervention and the frequency of visits are other factors which appear relevant in weight management interventions, as in adult participants they are inversely associated with cure abandonment [5].

The current orientation for improving adherence to treatments in pediatrics focuses on motivation, problem-solving skills, and reduction of post-treatment influence, resorting to a number of tools including web-based programs [6,7], “exergaming” [8,9,10], school interventions [11], summer camp [12], parent engagement [13,14,15], and automated mobile technology (MT) [16,17,18,19,20,21,22,23,24]. Studies focused on mobile phone interventions to improve lifestyle and preventing overweight and obesity are attracting interest in both the adult and pediatric population. In particular, the use of MT in children could be useful if utilised in innovative and complementary ways to traditional strategies (hybrid approach).

Differently from previous automated MT studies, our controlled study aims to evaluate the effectiveness of a personalized/non-automated mobile messaging intervention, with and without in-presence periodic recall visits inserted between regular visits, upon adherence to follow-up and improvement of anthropometric parameters and lifestyle.

## 2. Materials and Methods

### 2.1. Participants

Children aged 6–14 years old, affected by obesity (Body Mass Index (BMI) > 95th percentile for age and sex according to the Centers for Disease Control and Prevention (CDC) 2000 growth curves for 2–20 years old) [25] were recruited between January 2017 and February 2019 for this study developed in the Pediatric Obesity Clinic of our University Hospital.

In addition to the presence of obesity, inclusion criteria were consent to participate in the study and willingness to be contacted by phone (What’s App^®^) by the assigned coach, on their mobile phone or (if under 8 years old) via the mother.

### 2.2. Study Design

This quasi-experimental study consists of two comparative but separate phases: PediaFit 1.1 (January 2017–January 2018) and PediaFit 1.2 (January 2018–February 2019). In each of the two periods, patients at the end of the first visit were randomly allocated with 1:1 ratio to an Intervention (IG) or a Control group (CG) exclusively on the basis of the chronological order of their outpatient hospital booking. There were no patient transfers from one group to another.

The first pilot study (PediaFit 1.1) compared patients with standard treatment (CG) vs. patients who received standard treatment associated with a personalized messaging program (IG). In the second pilot study (PediaFit 1.2) monthly recall visits were added to the IG protocol. In particular, as illustrated in Figure 1 (panels A and B), patients in the Intervention groups received three regular visits (IG 1.1, *n* = 24) or seven in presence visits (IG 1.2, *n* = 30; three regular visits plus four recall visits). The 54 participants of both IGs received from their own coach (assigned during the first visit) three personalized messages/week during the intervals between their three-monthly regular visits. In addition, IG 1.2 had monthly in-presence recall visits which aimed to give better personalized support and reinforce the purpose of the weight and lifestyle management program.

Control groups (CG 1.1, *n* = 25; CG 1.2, *n* = 24; total *n* = 49) received standard treatment (indications for healthy nutrition and physical activity, plus regular visits every three months) without the MT messaging aid and without monthly recall visits.

Both groups received a first visit by a team consisting of a specialized pediatrician, a resident in pediatrics, a dietician and a medical student (IG coach). During the first and the other regular visits, the specialized pediatrician and the resident in pediatrics carried out anamnesis, physical examination and evaluation of the main anthropometric and lifestyle parameters, and the dietician evaluated the food diary. The patients were provided with a poster drawn up along the lines of the “food traffic light”, containing nutritional advice (Appendix B
Figure A1 and Figure A2, translated and original versions, respectively). After discussion of the poster, a visit to a dedicated Facebook^®^ page which is constantly updated was recommended [(https://www.facebook.com/Pediafit/(accessed on 10 February 2021)]. The designated coach (medical student) in charge of following the patient during the pilot project took care of the data collection and the messaging program throughout the study. The study was conducted according to the Helsinki Declaration [26] and approved by the Ethics Committee of the University of Salerno. Parents signed an informed consent with agreement to participate in the program, be contacted and allowing the use of the clinical data for research purposes.

### 2.3. Messaging Program and Recalls

Messages were sent to the children, if aged >8 years (*n* = 29), or to a parent, usually the mother (*n* = 25). Both programs lasted 24 weeks and each was divided into two parts: (a) self-monitoring and (b) challenge messages.

Self-monitoring messages (3 messages/week; Appendix C
Table A1 and Table A2), sent by the coach during the 12 weeks in the interval between the first and the follow-up regular visit, focused on healthy behavior (topics regarded sugary drinks, fruit and vegetables consumption, breakfast, portions, screen-time, physical activity, hours of sleep) accompanied by empathetic and personalized advice and/or encouragement. The child’s parent was required to submit body weight every weekend.

Challenge messages (3 messages/week; Appendix D
Table A3 and Table A4), were sent during the 12 weeks in the interval between the second and the third follow-up regular visit, in order to reinforce the healthy behaviors learned. The messages were preceded by an empathetic and personalized phone call from his/her coach as a reminder, along with a request for feedback.

PediaFit 1.2 (Figure 1, panel B) included, in addition, free intermediate monthly in presence visits (recall visits). Each patient in the PediaFit 1.2 IG underwent a total of four recall visits over six months (respectively at first, second, fourth and fifth month) during which the coach and the dietician recorded on site auxological parameters, current food diary, news about dietary lifestyle, physical activity, and any critical issues.

### 2.4. Data Collection

During baseline and follow-up regular visits (and also recall visits in the case of Pediafit 1.2) per protocol anthropometric parameters (body weight, height, waist circumference (CV), neck circumference (CC); BMI and BMI z-score), blood pressure, and obesity related acanthosis nigricans (AN) were measured. Standard laboratory and instrumental (abdomen ultrasound to search for hepatic steatosis) investigations were requested during the regular visit.

Lifestyle was investigated by requesting information (at baseline, during follow-up regular visits and also during recall visits only for PediaFit 1.2) about hours of sleep per night, minutes of physical activity per day/week, hours of sedentary lifestyle and screen-time per day, presence or absence of breakfast, number of meals per day, daily consumption of fruit and vegetable portions and calculated consumption of sugary drinks (mL) over the course of a week.

IG and CG data were collected at each regular and recall visit, and compared at the end of each study. At a later stage, the two studies were compared retrospectively.

### 2.5. Statistical Data Analysis

We carried out an exploratory evaluation of the distribution using a boxplot. Despite the wide variability of the individual data in some of the parameters, no consistent symmetry violations with particular outliers were identified. Continuous normally distributed parameters were reported as means ± standard deviation (SD). Dropout analysis was carried out by comparison of adherence to the follow-up of the two groups at the third and sixth month with Exact Fisher Test. For the analysis of anthropometric results and healthy behavior a T-Student test was performed, comparing the outcomes of compliant patients at the third and sixth month (the dropout at the sixth month in the CG was very high, therefore making the analysis poorly reliable). The analysis was intention-to-treat, without imputation of the missing data.

All analyses were performed using the Statistical Package for the Social Sciences (SPSS, version 17.02). Statistical significance was defined as *p* < 0.05 (two-tailed).

## 3. Results

The study included a sample of 103 patients (54 males (52%)], aged between 6 and 14 years (Mean = 10 years), allocated into PediaFit 1.1 [49 participants, 25 females (57%)), and PediaFit 1.2 (54 participants, 30 males (66%)). The baseline clinical parameters of all the participants are shown in Table 1.

Self-monitoring messaging received at least one feed-back by 100% of both the IG 1.1 and IG 1.2 participants, whereas messages with challenges received at least one feed-back by 67% and 96%, respectively. In the two messaging phases, compliance to messages (answers to more than 50% of the expected messages) was 75% and 81% in the IG 1.1, and 83% and 60% in the IG 1.2 group, respectively. Both the first and second recall visit were attended by 93% of IG 1.2 participants (dropout 7%). The third and fourth recall visits also reached comparable participation (90% and 93% of the IG 1.2, respectively).

As shown in Table 2, adherence to follow-up regular visits in general was better in IG vs CG and in IG2 vs IG1 both at the third and sixth month. The comparison between IG 1.1 and IG 1.2, showed that the latter had a statistically significantly higher adherence to follow-up at 6 months (3/30 = 10% vs. 15/24 = 62%; *p* 0.0001).

Anthropometric parameters at three months showed a statistically significant improvement in participants of IG1 compared to CG1, in particular for BMI (*p* = 0.026), BMI-zs (*p* = 0.018), percent reduction of WC excess (*p* = 0.02) and NC excess (*p* = 0.004). Other values also improved but did not reach statistical significance. At the sixth month assessment, the absolute values of parameters were still decreasing more in the IG1 vs. CG1 without statistically differences. The improvement of parameters at three months in the IG of PediaFit 1.2 compared to its CG was significant in particular for reduction of BMI (*p* = 0.04), BMI zs (*p* = 0.04), SBP (*p* = 0.02) and DBP (*p* = 0.02) values, and degree of AN (*p* = 0.00). The percentage of reduction of WC excess (*p* = 0.33) and NC (*p* = 0.30), although improved, did not reach statistical significance. At six months the comparison continued to show a significant improvement in BMI (*p* = 0.003) and AN degree (*p* = 0.0003) (Table 3).

Comparing the two phases of the study, it appears that at the three-month evaluation patients in the IG PediaFit 1.2 showed a greater reduction of BMI-zs (*p* = 0.01), excess WC% (*p* = 0.000) and degree of AN (*p* = 0.03) compared to patients in the IG PediaFit 1.1. The remaining anthropometric parameters also tended to improve more considerably in PediaFit 1.2 vs. PediaFit 1.1, but without reaching a statistically significant difference. Except for BMI zs and blood pressure, at the sixth month PediaFit 1.2 also performed better, without reaching statistical significance (Appendix A).

As depicted in Table 4, at the three month visit, improvement in lifestyle parameters of the PediaFit 1.1 IG vs. CG was statistically significant in particular for the reduction in the consumption of sugary drinks (*p* = 0.002), and the increase in daily fruit and vegetable consumption (*p* = 0.04). The other analyzed parameters, although improved in the IG, did not reach statistical significance. At six months there was no statistical significance for any of the parameters analyzed. In PediaFit 1.2, at three months the improvement in lifestyle parameters in the IG was statistically significant for all parameters assessed, except for the increase in sleep hours (but most children already had a standard of 8–9 h of sleep/night). At the sixth month evaluation, statistical significance was maintained only for fruits and vegetables consumption (*p* = 0.02) and screen-time (*p* = 0.04). The other analyzed parameters, although improved in the IG, did not reach statistical significance.

Comparison of the lifestyle changes in the IG PediaFit 1.2 and IG PediaFit1.1 (Appendix A) showed a statistically significant improvement in Physical Activity (*p* = 0.03 at three months and *p* = 0.01 at six months) and hours of sleep per night (*p* = 0.02 both at three and at six months). The other analyzed parameters also improved, but without reaching statistical significance.

## 4. Discussion

Communication technologies are an important part of children’s and adolescents’ lives and their use to encourage both positive lifestyle changes and adherence to care is an attractive and novel issue. In this study we report that a hybrid association of semi-personalized smartphone messaging plus monthly in-presence recall visits was associated with participants’ high feed-back rates to messages, adherence to follow-up regular visits, and significant improvements in obesity and lifestyle parameters over a 6 months period. Overall, the results were remarkably better than in a similar group who had no recall visits, and of controls who followed only the regular visits.

Adherence represents a crucial determinant of the success of a weight loss intervention [4]. Both arms of our intervention studies, especially the group with added recall visits, showed a noticeable low drop-out rate. Three pediatric systematic reviews [21,23,24] agreed on a lower tendency to dropout in participants in the mobile health technology assisted arms when compared to controls. In the most recent pilot study in adolescents with food addiction obesity included in a MT assisted intervention, adherence to face to face visits at six months’ follow-up reached 100% vs. 35% of a similar control group without MT [22], comparable to our own results.

Anthropometric and lifestyle indicators improved, however without reaching statistically significance vs. controls, in most of the literature cases [21,23,24]. In our study, instead, the above parameters of the two MT supported intervention groups showed a significant better outcome vs. their respective controls, and, as for adherence, the performance improved in those with added recall visits. Waist circumference and blood pressure, two important components of metabolic syndrome rarely considered in other MT assisted studies, tended to improve in parallel, mirroring the overall effectiveness of the measures.

Due to the hybrid nature of the interventions performed in the literature, without control groups it is not always easy to separate the merits of the MT vs. standard treatment. Some variables which cannot be objectified such as not-personalized contacts [23], coach’s empathy [22], messages doses, frequency [21,23] and content [23], may furthermore influence the outcome. This should be considered in the data interpretation and in the reproducibility of the study design. In our study a constant better performance of both MT assisted interventions could be appreciated vs. their own controls. In general, the addition of an arm with in presence free of charge recall visits resulted in a further improvement in the prefixed outcomes. This is in agreement with the idea that increasing the frequency of intervention visits is crucial for the management of childhood obesity [27]. This could imply that feedback from frequent visits may lead participants to feel themselves more monitored, as probably happened in our MT intervention group with added recall visits. The economic aspect is an important aspect which can influence the intervention outcome [22]; however, in our study we could not separate the appeal of saving money (i.e., the free of charge nature of the recall visits) from the higher intensity of intervention, an aspect which needs further consideration.

A number of other factors are likely to be accountable for some of the encouraging results obtained. We believe that the friendly nature of the MT use was appropriate to the purpose of the intervention. The study protocol with messages promoting healthy information tailored to the patients by a coach known by family and children met at their first visit, rather than by an automatic server, was likely to have facilitated the patient-doctor empathic alliance. Furthermore, the results of our study were likely to have been favored by the presence of a multidisciplinary team adequately trained in the management of obesity and its comorbidities, which may point toward the importance of an adequate medical training in obesity management. Together, these factors might have had possible additive effects.

Our pilot study has, however, a number of limitations which may have impacted or influenced the interpretation of the findings from our research, leaving some unanswered questions not adequately addressed. Firstly, the small size of the sample and, in some instances, the wideness of the standard deviation might have not been representative of the target population and not allowed an adequate statistical evaluation, respectively. On the other hand, the costs and/or time required by a dedicated coach vs. an automated approach might not be easily afforded for future studies with larger samples. Secondly, most lifestyle changes were based on self-reported evidence, which cannot be checked by the investigators. Thirdly, the short follow-up might not have allowed us to catch the process of lifestyle changes which, may require consolidation over years. Patient’s compliance may also have been positively or negatively influenced by the assigned coach’s empathy.

## 5. Conclusions

In presence and free of charge recall-visits may improve the prefixed outcomes of MT weight loss intervention programs. Future studies are needed to verify whether such an approach can conceivably also have cost/benefit efficacy in terms of the prevention of future obesity comorbidities.

## Figures and Tables

**Figure 1 nutrients-13-00631-f001:**
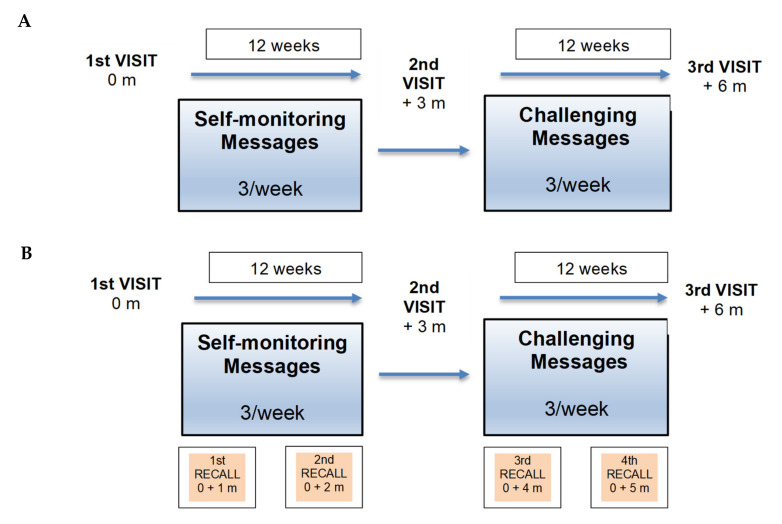
(Panel (**A**)) Intervention Group PediaFit 1.1 with three regular visits (VISITS) and three weekly messages. (Panel (**B**)) Intervention Group PediaFit 1.2 with three regular visits, three weekly messages, and four on site recall visits (RECALLS) for a total of 7 in presence contacts. (m = month).

**Table 1 nutrients-13-00631-t001:** Clinical parameters of the 103 patients allocated to Intervention and Control Groups of PediaFit 1.1 + PediaFit 1.2. at entry.

	Intervention Group Mean (SD)	Control Group Mean (SD)
Variable ^a^	First Visit *n* = 54 (24 M; age 9.7 years)	First Visit *n* = 49 (25 M; 10.4 years)
BMI (kg/m^2^)	29.2 (4.6)	30.4 (6.1)
BMI-zs	2.97 (0.5)	2.0 (0.8)
WC (cm)	85.1(10.2)	86.3 (19.3)
NC (cm)	33.1 (2.9)	33.5 (4.8)
SBP (mmHg)	115.2(13.2)	112.2 (15.2)
DBP (mmHg)	66.9 (12.2)	65.7 (12.0)
AN (grade)	1.5 (0.9)	1.33 (1.0)
F&V (portions/day)	1.2 (1.1)	1.5 (1.09)
SuD (mL/week)	894.3(514.0)	1441.6(1424.0)
Screen T (min/day)	199.4(110.0)	245 (126.9)
PA (min/week)	69.3(125.7)	80 (100.8)
Sleep (h/night)	8.2 (1.22)	7.8 (1.2)

**^a^** AN: Acanthosis Nigricans; BMI: body mass weight; BMI zs: z-score BMI; DBP: Diastolic Blood pressure; F&V: fruits and vegetables; NC: Neck circumference; PA: physical activity; SBP: Systolic blood pressure; Screen T: screen time; Sleep: hours of sleep per night. SuD: Sugary drinks; WC: waist circumference.

**Table 2 nutrients-13-00631-t002:** Adherence to follow-up.

Time	Compliant Patients N (%)								
	IG 1.1	CG 1.1	*p* Value	IG 1.2	CG 1.2	*p* Value	IG 1.1 + 1.2	CG 1.1 + 1.2	*p* Value
First visit (N)	24	25		30	24		54	49	
Regular visit at 3 Months	12 (50%)	6 (24%)	0.079	28 (93%)	7 (29%)	<0.0001	40 (74%)	13 (26%)	<0.0001
Regular visit at 6 Months	9 (38%)	2 (8%)	0.018	27 (90%)	3 (12%)	<0.0001	36 (66%)	5 (10%)	<0.0001
At least one check	14 (58%)	6 (24%)	0.021	28 (93%)	8 (33%)	0.00025	41 (75%)	16 (32%)	0.00001

IG = Intervention Group; CG = Control Group.

**Table 3 nutrients-13-00631-t003:** Changes in anthropometric parameters.

Variable	3 Months, Mean (SD)						6 Months, Mean (SD)					
Group	IG 1.1	CG 1.1	*p* Value	IG 1.2	CG 1.2	*p* Value	IG 1.1	CG 1.1	*p* Value	IG 1.2	CG 1.2	*p* Value
**n**	24	25		30	24		24	25		30	24	
BMI Kg/m^2^	−2.36 (1.29)	−0.94 (1.10)	0.026	−2.2 (0.9)	−1.18 (1.6)	0.04	−2.99 (2.96)	1 (0.42)	0.12	−4.6 (1.8)	+2.7 (2.8)	0.003
BMI zs	−0.28 (0.15)	−0.10 (0.11)	0.018	−1.29 (1.3)	−0.1 (0.2)	0.04	−0.33 (0.32)	0.06 (0.16)	0.14	−1.8 (0.7)	−0.2 (0.3)	0.2
Ex WC%	−36.11 (38.12)	3.20 (19.99)	0.02	−30.9 (23.83)	−20.83 (15.94)	0.33	−28.89 (43.65)	0 (0)	0.39	−34.19 (27.07)	−5.00 (7.07)	0.15
Ex NC%	−59.58 (42.20)	1.42 (25.80)	0.004	−38.41 (40.23)	−20.83 (18.00)	0.30	−54.03 (67.19)	21.25 (58.33)	0.199	−57.18 (44.52)	−12.50 (17.67)	0.18
SBP mmHg	−9.58 (9.87)	−5.00 (17.32)	0.453	−14.03 (8.5)	−3.5 (13.7)	0.02	−6.25 (14.33)	−5.00 (7.07)	0.911	−24.64 (25.79)	−3.5 (2.12)	0.27
DBP mmHg	−3.63 (7.10)	−1.25 (13.15)	0.92	−11.59 (15.36)	+5.2 (6.4)	0.02	−1.88 (10.67)	7.50 (10.61)	0.29	−2.37 (17.26)	−5.00 (0.0)	0.79
AN grade	−0.41 (0.51)	0.00 (0.63)	0.13	−0.8 (0.5)	+0.3 (0.5)	0.00	−0.75 (0.89)	0.00 (1.41)	0.36	−1.0 (0.6)	+1.0 (0.0)	0.0003

AN: Acanthosis Nigricans decrease; BMI: body mass index; BMI zs: z-score BMI; CG = Control Group; Ex WC: excess waist circumference by 95° percentile; Ex NC: Excess Neck circumference by 95° percentile; DBP: Diastolic Blood pressure; IG = Intervention Group; SBP: Systolic blood pressure.

**Table 4 nutrients-13-00631-t004:** Changes in lifestyle parameters.

Variable	3 Months, Mean SD						6 Months, Mean SD					
	IG 1.1 (*n* = 12)	CG1.1 (*n* = 6)	*p* Value	IG1.2 (*n* = 12)	CG1.2 (*n* = 6)	*p* Value	IG1.1 (*n* = 9)	CG1.1 (*n* = 2)	*p* Value	IG 1.2 (*n* = 9)	CG1.2 (*n* = 2)	*p* Value
SuD (mL/week)	−673.5 (487.5)	−57 (419.75)	0.002	−587.0 (367.8)	−35.7 (18.9)	0.02	−860.0 (586)	0.0 (424.3)	0.17	−718.0 (504.2)	−683. 3 (500.2)	0.67
ScreenT (min/die)	−45.0 (101.05)	6.67 (30.76)	0.38	−83.8 (93.0)	−8.5 (80.7)	0.02	−81.4 (95.9)	15 (63.6)	0.39	−118.7 (100.2)	20.0 (124.9)	0.04
Sleep (h/night)	0.3 (0.5)	0.0 (0.0)	0.22	0.6 (0.9)	0.85 (1.1)	0.55	−0.58 (1.65)	0.0 (0.0)	0.65	1.18 (1.5)	1.33 (1.52)	0.8
F&V (portion/die)	1.25 (1.09)	0.2 (0.4)	0.040	1.18 (1.6)	0.00 (0.5)	0.04	2.03 (1.2)	0.33 (0.57)	0.19	2.57 (1.1)	0.66 (1.15)	0.02
PA (min/week)	0.76 (12.55)	20 (36.17)	0.09	71.85 (118.0)	−30 (111.0)	0.03	11.2 (63.59)	15.0 (21.2)	0.59	112.2 (113.1)	133.3 (23.09)	0.7

CG = Control Group; F&V: fruits and vegetables; IG = Intervention Group; PA: physical activity; ScreenT: screen time; SuD: Sugary drinks.

## Data Availability

Data supporting the findings of this study are available from the corresponding author [P.V.] on request.

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
