# Peer review of "Healthy Lifestyle Management of Pediatric Obesity with a Hybrid System of Customized Mobile Technology: The PediaFit Pilot Project"

_nutrients, 2021, doi:10.3390/nu13020631_

Round 1
Reviewer 1 Report
The manuscript entitled “Healthy lifestyle management of pediatric obesity with a hybrid system of customized mobile technology: the PediaFit project.” presents interesting issue, but some areas must be corrected.
Major:
- The major limitation of the presented study is associated with very small studied groups (n = 24, 30, 49) with a serious drop outs, resulting in a very small final sample size of n = 5 for the control group (with n = 49 at the beginning). Such small studied group does not allow to conclude, as the obtained results are not representative for the general studied group. Taking this into account, the results are seriously biased and Authors are not able to conclude based on them.
- Authors presented the materials provided for the participants of the study, but they did not define why the materials are in English. If Authors provided materials in Italian, they should present both English and Italian version as a supplementary material. If the materials were provided in English, and being fluent English speaker was within the inclusion criteria, it should be indicated.
- Moreover, the manuscript is shabbily prepared and not prepared according to the instructions for Authors.
Abstract:
Authors should present the program precisely – any details are necessary here
Authors should present specific numeric values that were obtained and the results of the conducted statistical analysis (p-values).
Introduction:
Authors present a number of basic and even very trivial information that should not be presented in a scientific manuscript (e.g. “Prevalence of global childhood obesity increased noticeably in the past four decades (“globesity”).”) – Authors should be aware that they do not prepare the basic manual for students, or column of the newspaper, but a scientific paper that should be interesting for researchers from the area of food and nutritional sciences, so they should understand that their readers will have the nutritional knowledge.
Authors should present proper background for their study – as they studied body mass reduction program for children, they should address similar studies of similar programs and present challenges and problems which are defined by other authors (e.g. https://www.ncbi.nlm.nih.gov/pmc/articles/PMC5634063/; https://www.ncbi.nlm.nih.gov/pmc/articles/PMC7230971/; https://pubmed.ncbi.nlm.nih.gov/22995865/).
Authors should avoid multiple references that are not needed – it the current version of their manuscript even for very simple information (e.g. „low availability and suitability of the care system [10,11,14,15].”), Authors include numerous references (as for this phrase – four of them). The number of such references should be reduced – here and in other sentences.
Materials and Methods:
Authors should precisely describe the program with all the necessary details (e.g. how were measurements conducted, how was program developed, etc).
Authors did not indicate ho they verified normality of distribution and what procedure did they apply for non-parametrically distributed variables.
Results:
Authors should not include numerous sub-chapters, but rather precisely describe obtained results.
Discussion:
There is almost no discussion at all. Authors should discuss their results while compared with specific results of the studies by other authors.
Authors should not reproduce information presented in previous sections here.
Author Contributions:
It seems that contribution of some Authors was only minor and they did not participate in preparing manuscript. There is a serious risk of a guest authorship procedure which is forbidden. In such case they should be rather presented in Acknowledgements Section and not be indicated as authors of the study.
Author Response
REVIEWER 1 (YELLOW HIGLIGHTED CHANGES IN THE TEXT)
The manuscript entitled “Healthy lifestyle management of pediatric obesity with a hybrid system of customized mobile technology: the PediaFit project.” presents interesting issue, but some areas must be corrected.
Major:
1. The major limitation of the presented study is associated with very small studied groups (n = 24, 30, 49) with a serious drop outs, resulting in a very small final sample size of n = 5 for the control group (with n = 49 at the beginning). Such small studied group does not allow to conclude, as the obtained results are not representative for the general studied group. Taking this into account, the results are seriously biased and Authors are not able to conclude based on them.
R. Our study consists of a small sample and this drawback has further been acknowledged in the Limits of the study section. We hope that the encouraging results on a small sample will prompt new studies and insights based on our model approach.
The major dropout rate in the control group (standard treatment) is what happens in most obesity treatment programs, reflecting the main problem and cause of failure.
2. Authors presented the materials provided for the participants of the study, but they did not define why the materials are in English. If Authors provided materials in Italian, they should present both English and Italian version as a supplementary material. If the materials were provided in English, and being fluent English speaker was within the inclusion criteria, it should be indicated.
R. Attached please find the original materials which we translated for the reviewers and (hopefully) for the readership of Nutrients, should the MS be accepted and the Readers willing to use the same approach (Appendix A2 page 11; Appendix B2 page 13; Appendix C2 page 16)
3. Moreover, the manuscript is shabbily prepared and not prepared according to the instructions for Authors.
R. We have made attention to prepare a revised MS as much as possible close to the Journal instructions.
Abstract:
Authors should present the program precisely – any details are necessary here.
Authors should present specific numeric values that were obtained and the results of the conducted statistical analysis (p-values).
R. Within the possibilities of the Abstracts space limitations, now we have added
- a more precise description of the Program; lines 24,26
- and results plus statistics; lines 31,32,33
Moreover we prepared a Graphical Abstract as required by the Journal
Introduction:
1a Authors present a number of basic and even very trivial information that should not be presented in a scientific manuscript (e.g. “Prevalence of global childhood obesity increased noticeably in the past four decades (“globesity”).”) – Authors should be aware that they do not prepare the basic manual for students, or column of the newspaper, but a scientific paper that should be interesting for researchers from the area of food and nutritional sciences, so they should understand that their readers will have the nutritional knowledge.
R. we have removed basic/trivial information (lines 40-42)
(e.g. Prevalence of global childhood obesity increased noticeably in the past four decades (“globesity”). Excessive accumulation of body fat is a complex and multifactorial condition influenced by genetic heritage, eating habits, physical activity, in conjunction with environmental, psychological and social factors [1-3]. Excess weight in childhood is likely to persist also in adult age and is associated with an increased risk of developing chronic diseases and a series of conditions/comorbidities that reduce the quality of life
Childhood obesity is a major public health problem that increases the risk of medical comorbidities, health care costs and decreased quality of life. These factors underline the need for interventions to treat most overweight children and obesity. [1 4,5]. It is therefore necessary to develop effective interventions for adequate management [6].)
1b Authors should present proper background for their study – as they studied body mass reduction program for children, they should address similar studies of similar programs and present challenges and problems which are defined by other authors
R. Thank you for this further useful suggestion. In the revised version of the MS we have included more pertinent studies [see Bibliography ref n.3, 4,13-18, 21-23]
1c. Authors should avoid multiple references that are not needed – it the current version of their manuscript even for very simple information (e.g. „low availability and suitability of the care system [10,11,14,15].”), Authors include numerous references (as for this phrase – four of them). The number of such references should be reduced – here and in other sentences.
R. In the revised version of the MS we have removed throughout the text redundant references for the same subject.
Materials and Methods:
a. Authors should precisely describe the program with all the necessary details (e.g. how were measurements conducted, how was program developed, etc).
R. The revised version of the MS contains a better and detailed description of the Program (lines 71-79; 97-108)
b. Authors did not indicate how they verified normality of distribution and what procedure did they apply for non-parametrically distributed variables.
R. We carried out an exploratory evaluation of the distribution using a boxplot and no consistent symmetry violations with particular outliers were identified (despite the wide variability of the individual data)(lines 144-146).
Results:
Authors should not include numerous sub-chapters, but rather precisely describe obtained results.
R. We have now reduced the subchapters and simply indicated the results
Discussion:
There is almost no discussion at all. Authors should discuss their results while compared with specific results of the studies by other authors.
Authors should not reproduce information presented in previous sections here.
R. We have implemented the Discussion section and avoided to repeat previous concepts (lines 253-272; lines 294-303)
Author Contributions:
It seems that contribution of some Authors was only minor and they did not participate in preparing manuscript. There is a serious risk of a guest authorship procedure which is forbidden. In such case they should be rather presented in Acknowledgements Section and not be indicated as authors of the study.
R. We may assure the Reviewer that all the Authors participated to all the phases of the study and of the MS preparation (see more detailed Authors’ contribution) (lines 321-322)
Reviewer 2 Report
Lifestyle behavior studies aim to prevent childhood obesity are important. There are several key issues require authors’ attention: un-focus literature review, unclear description of randomization, unclear description of intervention and intervention dosage. The authors may consider using a different approach to test the intervention effect. There are excessive tables included in this ms and most of them can either combine or not necessary. Below, I provided feedback in a detail.
1.Introduction. The first paragraph-may focus on review the age group selected in the study. Currently, it is very broad. It looks like the high drop out rate is related to intervention taken place at medical center so please specify. It will be helpful to tell the readers the typical drop out because it is somewhat misleading to say affect up to 75%--this may only be reported by a few studies with challenge group not most studies.
2.2. Description of randomization is unclear. Did you randomize participants (1:1 ratio) to intervention or control group in year 2017 and the following year. If yes, it looks like it was a 2: 1 ratio. Or was this a cross over design? The second paragraph of 2.2. implies this was a RCT but it does not look like a RCT (somehow like quasi experiment design). Then the following paragraph says both intervention group became control groups. The study criteria is unclear. No description of recruitment. Did the study receive approval from any appropriate human subject review committee?
Figure 2 is challenging to follow. So the phase I intervention lasted 12 weeks and participants received 3 self-minoring message each week for the total of 12 week, then they attend an in person visit (why it says 0 +3 m-what does this mean). Also, what does recall mean? Clear lay out the intervention delivery and dose is critical so please revise.
I found description of intervention challenge to follow. It will be helpful to give readers some specific examples, instead of providing lots of very general and not helpful information. What does monthly recall visit meant? Did you ask participants to recall something? If yes, what were they asked and for what purpose?
2.4. it is unclear when the data were collect? Did you collect baseline and how did you collect data.
2.5. If this was a RCT why not apply intent to treat and linear model. Also, the analysis testing the intervention effect should control baseline measure.
- Result. This manuscript has excessive tables and most of them (> 8-9 tables) are not necessary. The intervention effect should be in one table but authors used several tables through the result sections.—which make reading and understanding tables challenge. Also, since the intervention and randomization were unclear described, it is difficult to follow the result sections. Since this may be an RCT, a CONSORT chart should be provided.
Author Response
REVIEWER 2 (GREEN HIGHLIGHTED CHANGES IN THE TEXT)
Lifestyle behavior studies aim to prevent childhood obesity are important.
There are several key issues require authors’ attention:
- un-focus literature review,
- unclear description of randomization,
- unclear description of intervention and intervention dosage.
- The authors may consider using a different approach to test the intervention effect.
- There are excessive tables included in this ms and most of them can either combine or not necessary. Below, I provided feedback in a detail.
1.Introduction.
a. The first paragraph-may focus on review the age group selected in the study. R The 1st paragraph of the revised MS has been modified with a more focused review of the age group
R. The revised bibliography focusing on the specific age group. [Ref.3,4: 13-18]
b. Currently, it is very broad. It looks like the high dropout rate is related to intervention taken place at medical center so please specify.
R. The drop-out rate issue has been better specified (line 43) [Ref 2]
c. It will be helpful to tell the readers the typical drop out because it is somewhat misleading to say affect up to 75%--this may only be reported by a few studies with challenge group not most studies.
R. The typical drop-out rate has been better discussed (lines 43-49).
2.2. Description of randomization is unclear.
a. Did you randomize participants (1:1 ratio) to intervention or control group in year 2017 and the following year. If yes, it looks like it was a 2: 1 ratio.
R. We randomized participants (1:1 ratio) to intervention or control group in year 2017 and made the same the following year. (line 78-89)
b. Or was this a cross over design?
R. It was not a cross over design
c. The second paragraph of 2.2. implies this was a RCT but it does not look like a RCT (somehow like quasi experiment design).
Then the following paragraph says both intervention group became control groups.
R. It is not an RCT, and IG did not become a CG. We agree it is an quasi-experiment design (lines 69; 75) .
d. The study criteria is unclear. No description of recruitment.
R. We have rewritten this section and corrected some misleading paragraphs (Lines 77-79)
e.Did the study receive approval from any appropriate human subject review committee?
R. Yes, the study received approval from the institutional university committee. (Line 109 -110 + see last page)
FIGURE 2 is challenging to follow.
a. So the phase I intervention lasted 12 weeks and participants received 3 self-minoring message each week for the total of 12 week, then they attend an in person visit (why it says 0 +3 m-what does this mean).
R: 0 +3 m- means: 3 months from time zero (first visit), We have now indicated this with “+3” months
b. Also, what does recall mean?
R : In our study Recall means refresher Visit/ Review visit (line 125-129). We indicate this now with the term Recall Visit
c. Clear lay out the intervention delivery and dose is critical so please revise.
R: Lay out of the intervention delivery and dose have been better specified (lines 93-95)
d. I found description of intervention challenge to follow.
It will be helpful to give readers some specific examples, instead of providing lots of very general and not helpful information.
R: we have removed very general infos and provided some specific examples
e. What does monthly recall visit mean? Did you ask participants to recall something? If yes, what were they asked and for what purpose?
R: see above answer b
2.4. it is unclear when the data were collect? Did you collect baseline and how did you collect data.
R. We have now specified better this aspect (lines130-142)
2.5. If this was a RCT why not apply intent to treat and linear model. Also, the analysis testing the intervention effect should control baseline measure.
R. It was not an RCT but rather a quasi-experiment design (see answer 2.2 c). The analysis was intention-to-treat, without imputation of the missing data. (lines151-152)
- Result.
This manuscript has excessive tables and most of them (> 8-9 tables) are not necessary.
The intervention effect should be in one table but authors used several tables through the result sections.which make reading and understanding tables challenge.
R. We have reduced the number of the Figures and of the Tables (we left some as online only supplemental files : page 18 Supplementary Tables 1 and 2) and we deleted another Table which was included in the Results Text of the manuscript (see Paragraphs 3.2 and 3,3)
4. Also, since the intervention and randomization were unclear described, it is difficult to follow the result sections. Since this may be an RCT, a CONSORT chart should be provided.
R.: our is not an RCT but rather a quasi-experiment design (see answer 2.2 c)
Round 2
Reviewer 1 Report
The manuscript entitled “Healthy lifestyle management of pediatric obesity with a hybrid system of customized mobile technology: the PediaFit project.” presents interesting issue, but some areas must be corrected.
Abstract:
Authors should present the program precisely – any details are necessary here
Results:
Authors should not include numerous sub-chapters, but rather precisely describe obtained results.
Discussion:
Authors should broaden their discussion. Authors should discuss their results while compared with specific results of the studies by other authors.
Author Response
Dear Reviewer # 1 - 2nd Revision ( LIGHT BLUE HIGHLIGHT)
Thank you very much for allowing us to improve further our manuscript.
We prepared a new-revision of the MS following your kind suggestions and requests.
Below please find the point by point reply.
===================================================
The manuscript entitled “Healthy lifestyle management of pediatric obesity with a hybrid system of customized mobile technology: the PediaFit project.” presents interesting issue, but some areas must be corrected.
Abstract:
1) Authors should present the program precisely – any details are necessary here
R) We added new details of the program in the Abstract
Results:
2) Authors should not include numerous sub-chapters, but rather precisely describe obtained results.
R. We eliminated sub-chapters and described obtained results more clearly, also by eliminating redundant values already shown in the Tables and Supplementary materials,
Discussion:
3) Authors should broaden their discussion. Authors should discuss their results while compared with specific results of the studies by other authors.
R. We thank the Reviewer for this suggestion. We partly rewrote the Discussion section according to his/her request, and included two systematic reviews and a most recent study (References # 22,23,24) to better compare our own results.